# Energy and environmental impacts of air-to-air heat pumps in a mid-latitude city

David Meyer [1] ✉, Robert Schoetter [2] & Maarten van Reeuwijk [1]

Heat pumps (HPs) have emerged as a key technology for reducing energy use and greenhouse gas emissions. This study evaluates the potential switch to air-to-air HPs (AAHPs) in Toulouse, France, where conventional space heating is split between electric and gas sources. In this context, we find that AAHPs reduce heating energy consumption by 57% to 76%, with electric heating energy consumption decreasing by 6% to 47%, resulting in virtually no local heating-related $CO_2$ emissions. We observe a slight reduction in near-surface air temperature of up to 0.5 °C during cold spells, attributable to a reduction in sensible heat flux, which is unlikely to compromise AAHPs operational efficiency. While Toulouse's heating energy mix facilitates large energy savings, electric energy consumption may increase in cities where gas or other fossil fuel sources prevail. Furthermore, as AAHPs efficiency varies with internal and external conditions, their impact on the electrical grid is more complex than conventional heating systems. The results underscore the importance of matching heating system transitions with sustainable electricity generation to maximize environmental benefits. The study highlights the intricate balance between technological advancements in heating and their broader environmental and policy implications, offering key insights for urban energy policy and sustainability efforts.

In 2021, residential space heating accounted for 17% of the final energy consumption in the European Union's group of 27[1] and 11% globally across the International Energy Agency's group of 61 countries, contributing to 8% of total carbon-dioxide ($CO_2$) emissions[2]. Heat pumps (HPs) represent a critical technological shift towards high-efficiency, electrically driven heating systems, offering a sustainable alternative to fossil fuel-based systems[3–5]. While previous research has underscored the potential of HPs in reducing buildings' energy consumption and greenhouse gas emissions[6,7], their effectiveness is contingent upon the system type, operational practices and electricity generation sources[7–9]. Previous analyses have leveraged a range of statistical, empirical, and numerical models to assess HPs' impacts on energy consumption and $CO_2$ emissions under various conditions[6,10–15]. Notably, ref. 6 show that in 53 out of 59 regions, HPs are less carbon-intensive compared to fossil fuel alternatives, even under current

electricity generation carbon intensities. This underscores the potential for HPs to reduce emissions, particularly in France, where the extensive use of nuclear power minimises their environmental impact. Conversely, ref. 10 illustrate a counterintuitive scenario in the United States, where switching to electric HPs could increase both heating costs and $CO_2$ emissions due to the predominant use of fossil fuels in electricity generation. This highlights that the environmental benefits of HPs are heavily dependent on the local energy mix. Similarly, ref. 11 provide empirical evidence from Arizona, challenging the assumption of universal energy savings from HP adoption. Their findings reveal no notable electricity savings during summer and an increased electricity demand in winter. Finally, ref. 15 explore the retrofitting of a UK dwelling with air-to-water HPs (AWHPs), noting a 12% reduction in $CO_2$ emissions but with a 10% increase in operational costs. Collectively, these studies underline the importance of considering regional energy

[1]Department of Civil and Environmental Engineering, Imperial College London, London, UK. [2]CNRM, Université de Toulouse, Météo-France, CNRS, Toulouse, France. ✉e-mail: d.meyer@imperial.ac.uk

sources and climate conditions in assessing HPs' environmental and economic impacts.

Physically, HPs move thermal energy from a cold location (e.g. outdoor air) to a warmer one[16] (e.g. indoor air). Depending on where the energy is sourced or released (e.g. in air, ground, or water) several HP types exist[3]. Air-to-air heat pumps (AAHPs; Supplementary Fig. 1) are the predominant type globally[17,18], meriting focused investigation into their performance and environmental impact. In simulating their behaviour, two variables are of main interest: capacity ($Q_{capacity}$), i.e. the amount of thermal energy ($Q_{heat}$) that an AAHP can provide at a given time of operation, and, work ($W$), i.e. the amount of electric energy used to drive the system at a given time of operation. The ratio of thermal energy moved into the building to that of electric energy used to drive the compressor defines the coefficient of performance (COP, i.e. COP = $Q_{heat}/W$)—a common metric to rate an AAHP's efficiency.

Here, we show through an integrated modelling approach, combining heating, ventilation and air conditioning (HVAC) models with building energy models (BEMs), urban canopy models (UCMs) and numerical weather prediction (NWP) models (e.g. refs. 19–22), the potential impact of AAHPs on energy consumption, urban climate dynamics, and $CO_2$ emissions in Toulouse, France. Similarly to previous meteorological studies investigating air conditioners' effects[23–29], and by assessing the COP in real-world scenarios, we contribute to the understanding of AAHP efficiency in diverse environmental settings offering insights into their transient behaviour under varying building and meteorological conditions. Through this detailed investigation, our research highlights the broader implications of AAHP deployment on electricity consumption and environmental sustainability, contributing comprehensive insights and offering a framework for future research and policymaking aimed at sustainable urban heating solutions.

## Results
### Building energy consumption
Using the offline SURFEX-TEB-MinimalDX models (Methods), we examine the impact of transitioning from fossil fuel and resistive heating-based systems to AAHPs on the annual heating and cooling energy consumption of Toulouse, France. This transition—assessed over the building energy consumption (BEC; i.e. considering all heating sources such as gas, electric, wood and oil; Supplementary Fig. 2a) and electric energy consumption (EEC) only—offers insights into the potential for energy savings and efficiency gains. The baseline scenario, based on and evaluated against results reported by ref. 30 (Supplementary Fig. 3), is compared with five distinct AAHP scenarios based on different values of rated coefficient of performance (RC). The baseline's annual BEC is 186 gigawatt day (GWd) for heating and 4.1 GWd for air conditioning (AC), which was almost non-existent in Toulouse in 2004[31]—this adds up to a total of 190 GWd (Fig. 1c and Table 1). Daily BEC for heating peaks at over 2 GWd during the cold spell at the end of January 2005 (Fig. 1b). BEC for cooling only occurs during warm spells in the summer. The median scenario, RC3.5, heralds a 72% reduction in BEC for heating, from 186 GWd in the baseline to 52 GWd (Table 1). The reduction spans from 61% in the least efficient AAHP scenario (RC2.5) to 78% in the most efficient scenario (RC4.5). For RC3.5, the cooling energy, while increasing by 54% to 6.3 GWd to reflect additional load from using AAHPs for cooling in the summer (rebound effect–Methods), still results in a net energy consumption reduction of 69% to 59 GWd for the combined heating and cooling. The timeseries of the differences of daily building energy consumption between baseline and RC scenarios (Fig. 1a) shows that the reduction in BEC is largest (above 1.25 GWd) during the cold spell of January 2005. Cooling energy consumption rises in all RC scenarios due to the exclusive use of electrically driven AC systems, but this is offset by a much larger reduction in heating energy consumption in the winter.

### Electric energy consumption in the centre of Toulouse
While BEC captures the energy buildings consume from all fuel sources, transitioning to AAHPs removes the need for all but electrical sources. Therefore, only the electric energy consumption (EEC) is investigated hereafter. The energy disaggregation method is validated against a prior assessment by ref. 32, focusing on Toulouse's centre (Fig. 2a–c and Table 1) where evaluation data are present, and found to be comparable (Supplementary Fig. 4a, b). With electric resistive heaters accounting for 59% of the heating in Toulouse's centre (Supplementary Fig. 2b), in the median scenario, RC3.5, electric energy

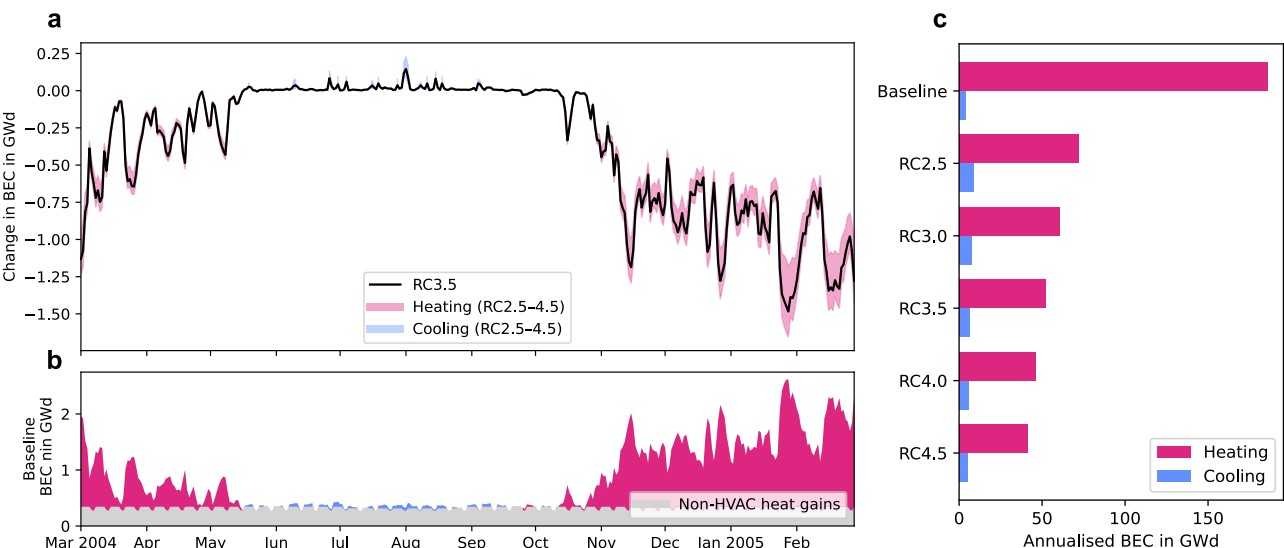

**Fig. 1 | Heating and cooling building energy consumption in the Toulouse agglomeration. a** Timeseries of daily averaged change in building energy consumption (BEC) in the Toulouse agglomeration from baseline in gigawatt day (GWd) showing the median rated coefficient of performance (RC) scenario (RC3.5, solid black line) and heating (pink) and cooling (blue) range for RC between 2.5 and 4.5 (shaded area), **b** baseline daily averaged BEC broken down into heating (pink), cooling (blue) and non-HVAC (heating, ventilation and air conditioning) heat gains such as lighting, electrical appliances and domestic warm water (grey; mean: 0.32 GWd, $n$ = 365 samples) components, and **c** annualised BEC for heating (pink) and cooling (blue).

**Table 1 | Annual heating and cooling energy consumption**

| | Heating | | | Cooling | | Heating and cooling | | |
|---|---|---|---|---|---|---|---|---|
| | BEC | EEC$_{Centre}$ | EEC | BEC/EEC* | EEC$_{Centre}$ | BEC | EEC$_{Centre}$ | EEC |
| | GWd (%) | GWd (%) | GWd (%) | GWd (%) | GWd (%) | GWd (%) | GWd (%) | GWd (%) |
| Baseline | 186 | 2.4 | 82 | 4.1 | 0.31 | 190 | 2.7 | 86 |
| RC2.5 | 72 (−61) | 1.9 (−24) | 72 (−12) | 8.9 (+115) | 0.50 (+64) | 81 (−57) | 2.4 (−14) | 81 (−6) |
| RC3.0 | 61 (−67) | 1.6 (−36) | 61 (−26) | 7.4 (+79) | 0.42 (+37) | 68 (−64) | 2.0 (−28) | 68 (−21) |
| RC3.5 | 52 (−72) | 1.3 (−45) | 52 (−36) | 6.3 (+54) | 0.36 (+17) | 59 (−69) | 1.7 (−38) | 59 (−32) |
| RC4.0 | 46 (−75) | 1.2 (−52) | 46 (−44) | 5.5 (+34) | 0.31 (+2) | 52 (−73) | 1.5 (−46) | 52 (−40) |
| RC4.5 | 41 (−78) | 1.1 (−57) | 41 (−50) | 4.9 (+19) | 0.28 (−9) | 46 (−76) | 1.3 (−52) | 46 (−47) |

Results are grouped for the building energy consumption (BEC) in gigawatt day (GWd) over the entire simulation domain (Fig. 1c), the electric energy consumption over the centre of Toulouse (EEC$_{Centre}$; Fig. 2c), and the electric energy consumption over the entire simulation domain (EEC; Fig. 3c).
*Same as space cooling as this is always generated by electrically driven air-conditioning systems.

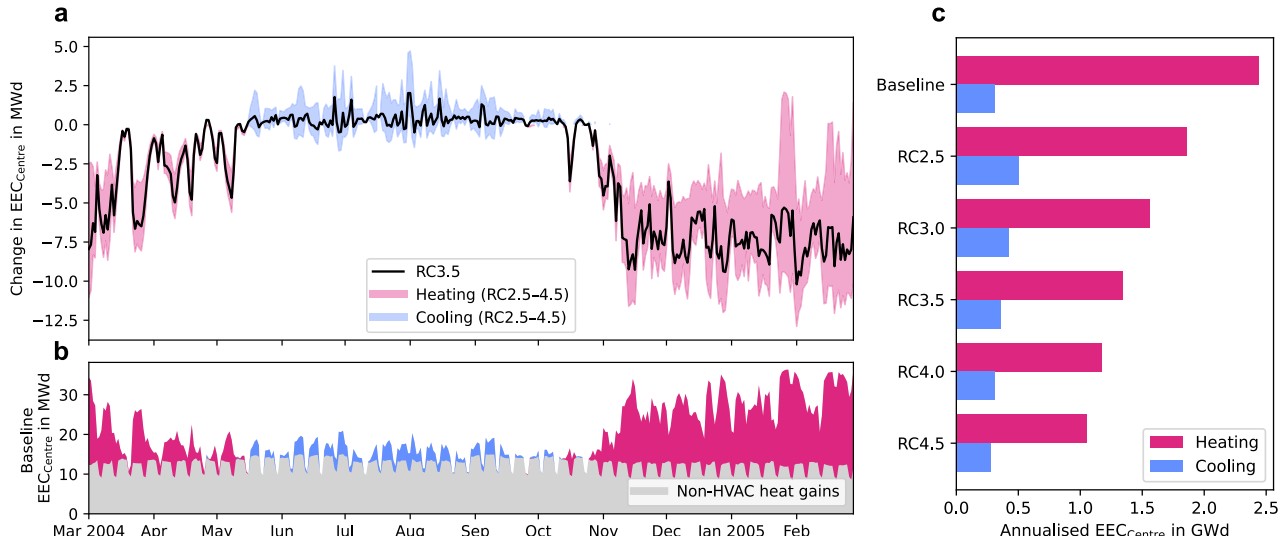

**Fig. 2 | Heating and cooling electric energy consumption in the centre of Toulouse. a** Timeseries of daily averaged change in electric energy consumption in the centre of Toulouse (EEC$_{Centre}$; as shown in Fig. 6c's shaded area) from baseline in megawatt day (MWd) showing the median rated coefficient of performance (RC) scenario (RC3.5, solid black line) and heating (pink) and cooling (blue) range for RC between 2.5 and 4.5 (shaded area), **b** baseline daily averaged EEC$_{Centre}$ broken down into heating (pink), cooling (blue) and non-HVAC (heating, ventilation, and air conditioning) heat gains such as lighting, electrical appliances, and domestic warm water (grey; mean: 12.39 MWd, $n = 365$ samples) components and **c** annualised EEC$_{Centre}$ in gigawatt day (GWd) for heating (pink) and cooling (blue).

consumption for heating in the city centre is reduced by 45%, from 2.4 GWd in the baseline to 1.3 GWd. This reduction is part of a broader trend observed across all scenarios, where the shift to AAHPs decreases electric heating energy consumption by 24 to 57% (Table 1). Similarly, as for BEC, the cooling energy consumption increases because of rebound effects by 17% to 0.36 GWd for RC3.5. The timeseries of differences in daily EEC between RC3.5 and the Baseline shows strong fluctuations (Fig. 2a). These are influenced not just by temperature-dependent energy consumption, as seen with conventional heating methods, but also by the dependency of the COP on both external and internal air temperature and humidity. For the RC3.5 scenario, the actual COP falls below 2.5 on multiple occasions during the coldest winter days (Supplementary Fig. 5).

### Electric energy consumption

Under the assumption that the modelling technique and energy disaggregation are valid, the analysis extends to the full simulation domain (EEC). Here, electric resistive heaters are responsible for 52% of Toulouse's heating energy consumption (Supplementary Fig. 2a), projecting a baseline electric energy consumption decrease of 36% to 52 GWd for heating but an increase of 54% to 6.3 GWd for cooling for

RC3.5 (Fig. 3a–c and Table 1). This increase in cooling energy consumption due to rebound effects highlights the dual role of electric energy in heating and cooling and points to the increased reliance on electrically driven systems for space cooling. Notably, even with substantial differences in building and heating characteristics between the central and outer regions (Supplementary Fig. 6), the EEC across the entire simulation domain mirrors the trends observed for BEC and EEC$_{Centre}$. The RC heating scenarios show the broader electricity savings potential, with reductions ranging from 12% in the least efficient to 50% in the most efficient. The pattern of electricity savings extends to the combined heating and cooling energy consumption, with this median RC3.5 scenario showing a 32% reduction in total EEC. As for BEC, the cooling electric energy consumption exhibits an increase in all scenarios due to the exclusive use of electrically driven AC systems.

### Surface energy balance

In the context of urban climate studies, sensible heat flux plays a crucial role in the urban surface energy balance, reflecting the impact of human activities and natural processes on city temperatures. This is particularly evident in the winter-averaged sensible and anthropogenic heat flux densities (Fig. 4). The analysis employs the offline SURFEX-

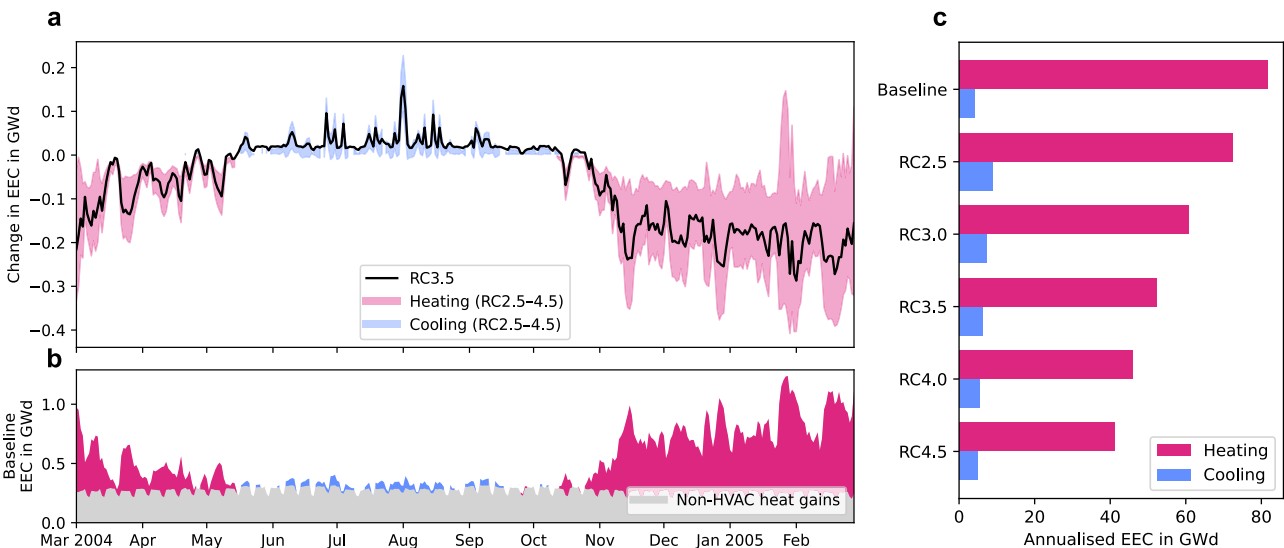

**Fig. 3 | Heating and cooling electric energy consumption in the Toulouse agglomeration. a** Timeseries of daily averaged change in electric energy consumption (EEC) in the Toulouse agglomeration from baseline in gigawatt day (GWd) showing the median rated coefficient of performance (RC) scenario (RC3.5, solid black line) and heating (pink) and cooling (blue) range (shaded area), **b** baseline daily averaged EEC broken down into heating (pink), cooling (blue) and non-HVAC (heating, ventilation, and air conditioning) heat gains such as lighting, electrical appliances, and domestic warm water (grey; mean: 0.27 GWd, $n = 365$ samples) components and **c** annualised EEC for heating (pink) and cooling (blue).

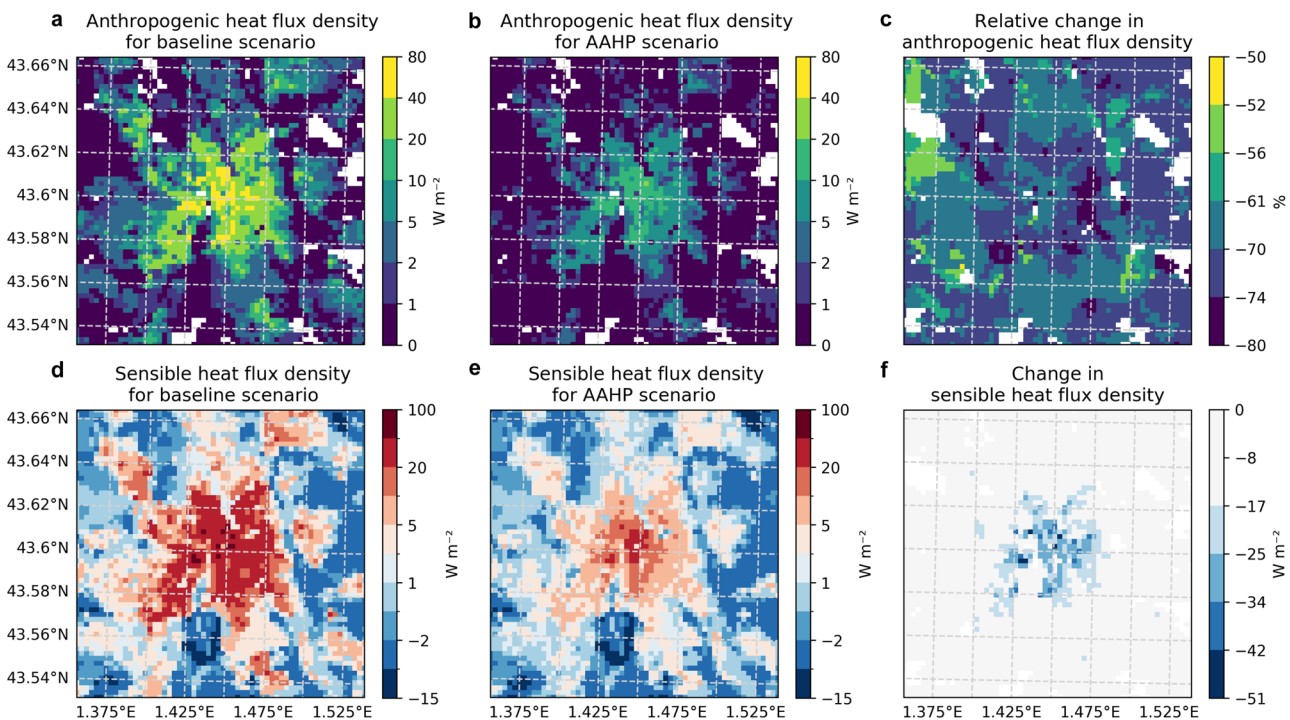

**Fig. 4 | Spatial distribution of average anthropogenic and sensible heat fluxes for the winter (December, January, February) for the rated coefficient of performance 3.5 scenario (RC3.5). a** Anthropogenic heat flux due to building energy consumption (with heating systems in place in 2004 as natural gas (Supplementary Fig. 6d), electricity (Supplementary Fig. 6c), wood (Supplementary Fig. 6e), oil (Supplementary Fig. 6f), **b** same as (**a**), but with all heating systems replaced by AAHPs and (**c**) relative difference between (**b**, **a**). **d**, **e**: same as (**a**, **b**), but for the sensible heat flux. **f** absolute difference between (**e**, **d**).

TEB-MinimalDX models to compare the baseline scenario with the RC3.5 scenario, excluding heat contributions from traffic and industry. The anthropogenic heat flux, stemming solely from building energy consumption, peaks at up to 80 W m$^{-2}$ in Toulouse's centre. This is attributed to dense building areas characterised by a high heated floor area (plane area building density above 0.5 and 4–5 storey buildings;

Supplementary Fig. 6a, b), and the low insulation of traditional buildings. The anthropogenic heat flux is reduced by about 50 to 80% in the AAHP scenario (Fig. 4c) given the superior efficiency of AAHPs compared to conventional heating systems. Figure 4d, e show the average winter sensible heat flux under both baseline and AAHP scenarios. During winter, when solar radiation is low, the sensible heat flux's

spatial distribution closely aligns with that of the anthropogenic heat flux. In urban areas, particularly the city centre, the baseline scenario records sensible heat flux values as high as 100 W m⁻² contrasting with slightly negative values of about −15 W m⁻² in rural areas. The shift to AAHPs reduces the sensible heat flux by about 50 W m⁻² in densely urbanised areas (Fig. 4f), highlighting its importance in the urban surface energy balance and its sensitivity to changes in heating technologies.

## Near-surface air temperature

Using the two-way coupled MesoNH-SURFEX-TEB-MinimalDX model, the effect of AAHPs on the 2-m air temperature at 17 distinct locations (Figs. 5a, 6b) is investigated during the cold spell between 22 and 30 January 2005 (Fig. 6d) focusing on the differential impact between the baseline and the RC3.5 scenario. A comparison of surface heat flux density reveals consistent findings between offline and online model versions (Supplementary Fig. 7). On average, the deployment of AAHPs reduces the 2-metre air temperature by less than half a degree Celsius (Fig. 5a). This effect arises because AAHPs efficiently transfer heat from

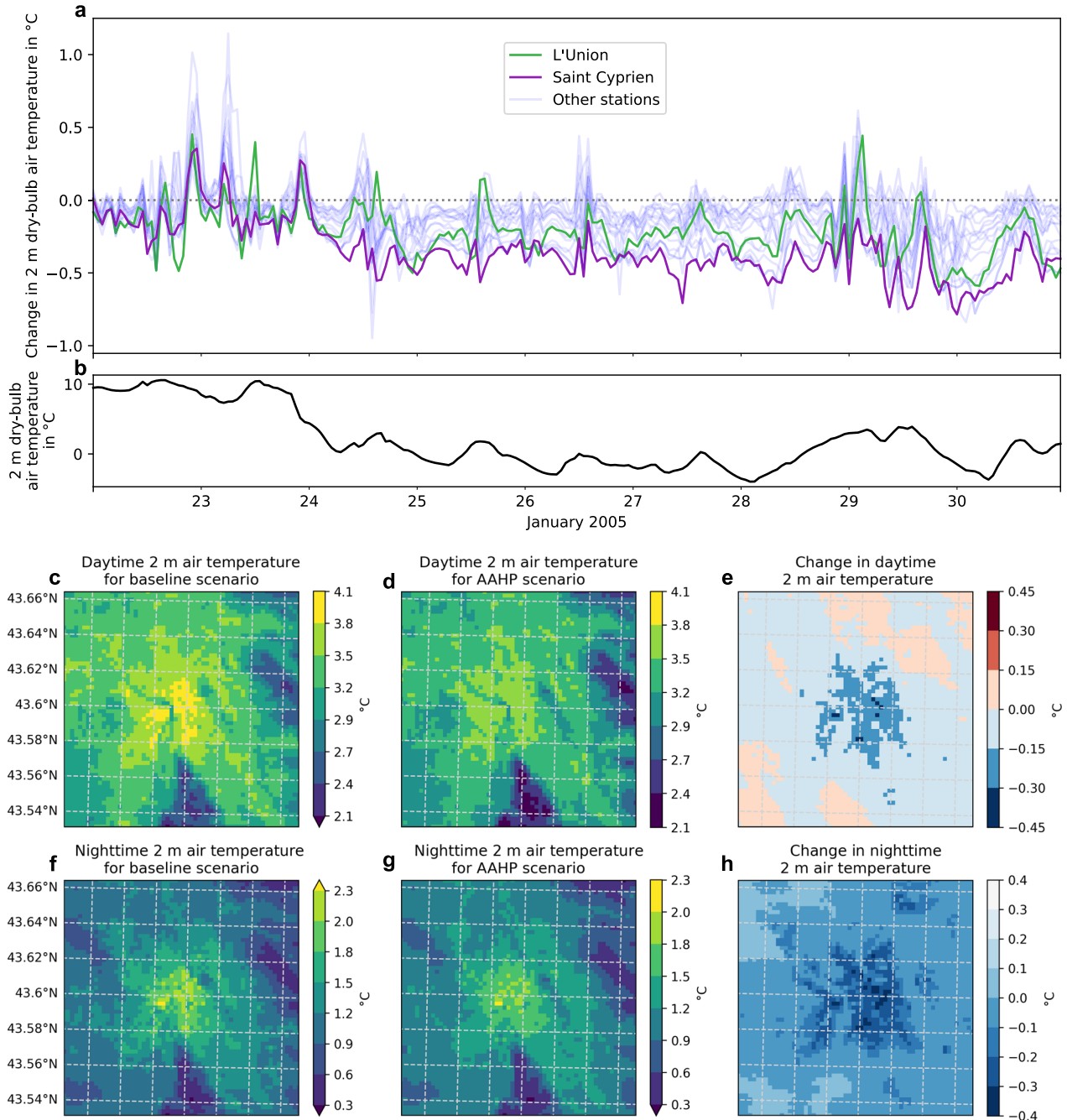

**Fig. 5 | Timeseries of the effect of AAHPs on near-surface air temperature during a cold spell from 24 January to 30 January 2005. a** Change in 2 m dry-bulb air temperature at the 17 observational stations with line at 0 °C shown with grey dotted line, **b** average 2 m air temperature at the 17 observational stations, **c–e** comparison of daytime (10:00–16:00 local time) 2 m air temperature for baseline (**c**) and AAHP (**d**) scenarios, and **f–h** comparison of nighttime (00:00–06:00 local time) 2 m air temperature for **f** baseline and **g** AAHP scenarios. Differences between baseline and AAHP are illustrated for **e** daytime and **h** nighttime. A comparison of near-surface air temperature for each station is made in Supplementary Fig. 8.

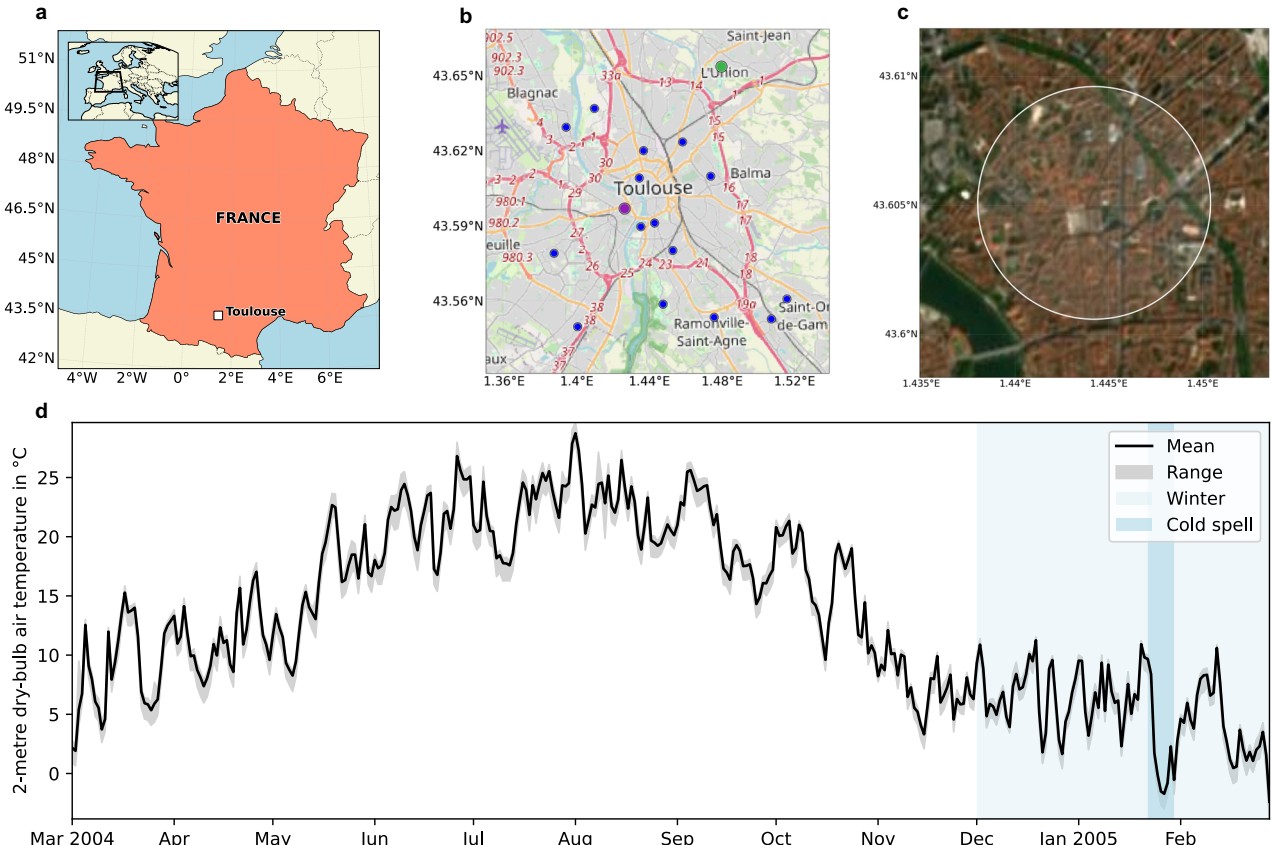

**Fig. 6 | Investigation domain and time used in the simulations. a** The location of Toulouse is shown relative to France and mainland Europe (inset). **b** The simulation domain with 17 weather stations is shown by blue circles; the larger purple and green circles show the stations closest to (Saint Cyprien) and furthest away from (L'Union) the centre of Toulouse. **c** The domain for the centre of Toulouse used to evaluate the electric energy consumption is shown with a white circle centred on the city's main square. **d** Observed daily averaged 2 m air temperature from the 17 weather stations is shown for the simulation period between March 2004 and February 2005. Offline simulations are conducted for the winter period from December 2004 to February 2005. Online simulations are conducted for the cold spell between 22 and 30 January 2005 to estimate the impact of AAHPs on the local microclimate. Copyright and licence: **a** Natural Earth/ Public domain; **b** OpenStreetMap Foundation/Open Data Commons Open Database License; **c** Sentinel-2 cloudless https://s2maps.eu by EOX IT Services GmbH (Contains modified Copernicus Sentinel data 2020)/Creative Commons Attribution-NonCommercial-ShareAlike 4.0 International.

the outdoor environment to indoor spaces, thereby emitting less heat back into the atmosphere compared to conventional fossil fuel-based heating systems, such as gas or electric heaters (Supplementary Fig. 1). Notably, the urban site of Saint Cyprien (Fig. 6b, purple) exhibits a larger temperature reduction (Fig. 5a purple and Supplementary Fig. 8q) than peripheral stations such as L'Union (Figs. 6b, 5a green and Supplementary Fig. 8a). This discrepancy is attributed to the lesser variation in heating energy consumption and sensible heat fluxes in remote areas (Fig. 5c, d, f, g). In the daytime and nighttime averages (Fig. 5e, h), the 2-m air temperature reduction does not exceed 0.45 °C, with the largest reductions observed in the centre.

## Discussion

Findings reveal a reduction in annual BEC and EEC by 57 to 76% and by 6 to 47%, respectively, based on RC values ranging from 2.5 to 4.5. Although this reduction shows the high efficiency of AAHPs—underscoring the potential of electric heating solutions to lower urban energy demands in the heating sector—this transition may not universally apply, especially in cities reliant on fossil fuel-based heating, where a switch to AAHPs could increase overall electric energy consumption[10,11]. Crucially, the electrification of heating through AAHPs underscores the need for sustainable electricity generation and highlights the challenges to HP's efficiency. The efficiency of AAHPs, as represented by the COP, inherently decreases as the external temperature drops, introducing a second-order dependency on

temperature. This efficiency decline during colder periods would thus require a careful approach to energy planning and grid management, emphasising the need for a robust energy integration to prevent overload and ensure stability. The example of Toulouse demonstrates the potential of AAHPs to enhance energy efficiency and reduce reliance on electric heating in environments predominantly heated by resistive elements. Yet, generalising these results to diverse urban contexts requires a deep understanding of local energy landscapes, heating practices, and the intricate balance between temperature, AAHP's efficiency, and energy consumption. As urban centres evolve, integrating AAHPs into their energy strategies offers a promising pathway towards sustainable heating. However, it also compels a comprehensive strategy to navigate the infrastructural and technological intricacies posed by the dual challenges of electrification and the inherent efficiency variability of AAHPs in response to cold weather conditions.

The adoption of AAHPs in Toulouse improves urban environmental sustainability, notably by reducing $CO_2$ emissions. Transitioning to AAHPs represents a crucial strategy in climate change mitigation, offering large reductions in $CO_2$ emissions within urban settings. This shift is particularly impactful in Toulouse, where the prevalent use of low-carbon electricity, primarily from nuclear sources (70.6% in France in 2019[33]), facilitates a decrease in $CO_2$ emissions through the transition to all-electric heating systems. For the baseline scenario, the local $CO_2$ emissions due to heating are up to 40 kg $CO_2$

$m^{-2}$ in the centre (Supplementary Fig. 9a). For the Toulouse agglomeration, these are zero during the summer, but they can reach up to 7000 t $CO_2$ day$^{-1}$ during cold spells in winter (Supplementary Fig. 9b) however, if the electricity consumed by AAHPs were generated using fossil fuels, indirect $CO_2$ emissions outside the city would be greater than those produced locally by combustion boilers. These findings indicate that, while direct emissions in urban areas decrease, assessing the broader implications of electricity generation's carbon intensity is essential. Notably, during winter, the potential for reducing local $CO_2$ emissions is large, underscoring the importance of aligning energy sourcing with sustainable practices to maximise environmental benefits. Furthermore, the study contributes to the understanding of AAHPs' impact on urban climate, particularly regarding surface energy balance and near-surface air temperatures. The analysis reveals that AAHPs can modestly alter urban microclimates, evidenced by slight reductions in near-surface air temperature of about 0.5 °C during the cold spell. Although these changes are small and expected to dissipate quickly due to prevailing wind conditions, further minimising their impact on system efficiency and thus unlikely to meaningfully alter AAHPs' performance through feedback, they underscore the complex interaction between heating technologies and urban climate dynamics.

Integrating AAHPs presents notable challenges, particularly regarding the rebound effect and the environmental impact of refrigerant leaks. The rebound effect states that as energy efficiency increases, so does energy consumption due to the reduced cost of usage[34]. In terms of heating and air conditioning, if the efficiency of these systems improves, occupants might use them more often, thus offsetting energy savings. Quantitatively, the rebound effect has been estimated to be about 10–30% for home heating[35]. Ref. [36] found a substantial rebound effect (38 to 86%) for residential electricity consumption in France, but they did not focus on HPs. Ref. [37] found a rebound effect of 20% for AAHPs in Danish residential buildings, due to a higher design temperature, a longer heating period, a larger heated floor area, and the potential use of the installed HPs as ACs. Ref. [38] found evidence for a potential rebound effect for residential buildings in the UK since, currently a part of the households are restricting their energy consumption due to the high cost of energy. Furthermore, households who install HPs might use them as ACs in the summer, and as the climate warms, occupants may start using air conditioning even if they have not had it in the past. In this study, an increase in energy consumption for cooling of about 54% to 6.3 GWd is estimated during the summer; however, this may increase considerably in future climates.

The environmental repercussions of refrigerant leaks with high global warming potential (GWP) warrant consideration. Difluoromethane (R32)—the most popular refrigerant used in today's residential HVAC systems[39]—with a GWP 675 times that of $CO_2$, poses risks to climate goals despite the efficiency gains of AAHPs. As such, two potential issues can be identified: one due to leaks and the other due to end-of-life recycling. For the former, the Intergovernmental Panel on Climate Change (IPCC) estimates that, in developed countries, 1% of an HP's refrigerant is lost to the environment. Therefore—assuming a total AAHP's heating capacity of 6.64 GW and an average charge of 0.25 kg $CO_2$ equivalent (CO$_2$e) per kW capacity[40]—the refrigerant lost to the environment is 11,205 t CO$_2$e year$^{-1}$, or 31 t CO$_2$e day$^{-1}$. Compared to the current baseline estimate of 577,740 t $CO_2$ year$^{-1}$ from March 2004 to February 2005, or about 1,600 t $CO_2$ day$^{-1}$, these estimates remain far below the baseline estimate. For the latter issue about end-of-life recycling, the IPCC estimates that the refrigerant from a fifth of all HPs is released into the atmosphere rather than being recycled. With this, 224,100 t CO$_2$e would be released at the end of the HPs lifetime—less than half of the 577,740 t $CO_2$ released annually (March 2004 to February 2005) in the baseline scenario.

The effective deployment of AAHPs requires careful consideration of a range of factors, beyond just practical constraints. These include aesthetic concerns, acoustic emissions, and adherence to urban planning regulations, which are particularly pertinent in areas with high population densities or historical importance. Furthermore, the efficiency of AAHPs in cold spells may present additional challenges, highlighting the need for thoughtful implementation strategies. Despite AAHPs maintaining an average coefficient of performance (COP) of about 3 even at 0 °C, as revealed by a recent meta-analysis[41], fluctuations in indoor and outdoor temperatures can still impact their transient performance. This variability underscores the complexities of energy management and electrical grid stability, further emphasising the importance of addressing both the initial adoption challenges and the operational considerations that affect the long-term success and sustainability of global AAHP deployments. In Toulouse, recent data from the Occitania administrative region[42], reveal a surge in HP installations, totalling 186,750 units in 2021 (82% AAHP, 18% AWHP). Notably, 94% of these occurred within residential settings, with 77% representing new installations rather than replacement of existing units. Considering that Occitania's household count stands at approximately 2.7 million, this surge implies that 5% of households adopted HPs in 2021 alone, reflecting the broader transition towards energy efficiency and reduced carbon footprints in Toulouse's residential settings.

In conclusion, the deployment of AAHPs in Toulouse, underscores AAHPs' potential to reduce energy consumption and contribute to urban sustainability goals. Our analysis highlights the critical role of the existing heating infrastructure in realising these energy savings. As such, while the shift from a majority-share resistive heating mix to AAHPs represents a clear path toward efficiency, transitioning from other fossil fuel-based sources such as gas boilers requires a deep understanding of the trade-offs between immediate energy savings and long-term environmental benefits. As cities strive for greater sustainability, the findings from Toulouse offer valuable insights into the strategic implementation of AAHPs to balance energy efficiency with environmental priorities. Policymakers and urban planners are thus encouraged to consider the local context in deploying HP technologies, ensuring that the move towards electrification aligns with both regional energy profiles and broader climate objectives.

## Methods
### Numerical urban climate model
The Town Energy Balance (TEB) model[43] is a physics-based UCM used to calculate the exchange of momentum, energy, and water between cities and the atmosphere. TEB is available as standalone software[44] or as a component in the EXternalised SURFace SURFEX[45]. Additionally, it is available in numerical weather forecasting and research platforms such as Meso-NH[46] and WRF-TEB[22] through WRF-CMake[47]. In this study, SURFEX-TEB version 8.2[30,48] is used. It computes the urban surface energy budget (net radiation, turbulent sensible and latent heat flux, and storage heat flux) as a function of meteorological forcing (air temperature and humidity, wind speed, downwelling solar and terrestrial radiation, and precipitation rate) by assuming a simplified street canyon geometry. The energy budget of a representative building at a district scale (e.g. 250 m × 250 m) is solved to compute the indoor air temperature, humidity, and building energy consumption as a function of simulated meteorological conditions, physical characteristics of the building envelope (e.g. roof and wall materials, windows), and human behaviour (e.g. design temperature for heating and air conditioning)[30,32,49]. A surface boundary layer scheme[50] is used to calculate vertical profiles of the meteorological variables in the urban roughness sublayer. The model allows to simulate realistic values of outdoor and indoor air temperature and humidity and use them as boundary conditions to the AAHP model.

## Numerical AAHP model

To simulate AAHPs, the HVAC model MinimalDX version 0.3.0[51]—a simplified single-speed, direct-expansion (DX) coil model from Energy Plus[52]—is coupled to SURFEX-TEB. MinimalDX allows the investigation of a change in performance as a function of indoor and outdoor conditions using bivariate quadratic fits to model the capacity and the electric input ratio (EIR = 1/COP) as a function of indoor wet-bulb temperature and outdoor dry-bulb temperature. These curves are either generated for a specific HP or AC model and make, or for more general classes (e.g. domestic or residential split type AC unit)[53]. MinimalDX is configured with performance and capacity curves from ref. 53. In this study, five scenarios of rated COP (RC) values of 2.5, 3.0, 3.5, 4.0 and 4.5 are investigated to reflect typical ranges of HP performance in mild to cold climates[41]. In modelling HPs, the influence of internal fans on energy consumption or heat gains is not considered due to their large variability between systems and their overall small impact. To account for defrost operation, a resistive defrost strategy at 20% of the total rated capacity when the outdoor dry-bulb air temperature is below 0 °C is assumed.

## Simulated AAHP scenarios

The baseline is defined the same as ref. 30: a detailed simulation of building energy consumption for Toulouse between March 2004 to February 2005. It is chosen as (1) it offers a high-resolution snapshot of building energy consumption in Toulouse, both spatially (at 250 m) and temporally (daily) and (2) it has undergone meticulous scrutiny and evaluation, ensuring its reliability. This baseline serves as a reference model for the heating systems present in Toulouse during the 2004–2005 timeframe, with all variables and configurations precisely mirroring those detailed in ref. 30. The analysis then encompasses five distinct AAHP scenarios, each representing a varying rated COP (RC), denoting the COP at standard conditions as specified by the manufacturers. While these RC values, set at 2.5, 3.0, 3.5, 4.0 and 4.5 serve as a reference, MinimalDX dynamically adjusts the COP in response to varying conditions like air temperature. For each scenario, an independent simulation is performed, which is subsequently compared to the baseline. To estimate the impact of AAHP on energy consumption, offline SURFEX-TEB simulations are made—i.e. forced with observed meteorological data from the centre of Toulouse. During the summer, AAHPs are repurposed as ACs. Lastly, to consider the impact of AAHPs on the microclimate, SURFEX-TEB is coupled with an atmospheric modelling framework to simulate a cold spell between 22 and 30 January 2005.

## Design temperature for heating and air conditioning

The heating design temperature in occupied (and vacant, where applicable) buildings for baseline and RC scenarios as a function of building use and heating system's efficiency is: 18.6 °C (17.6 °C) and 22.0 °C for single-family high and low-efficiency homes, respectively; 18.4 °C (17.4 °C) and 22.3 °C for multi-family high and low-efficiency homes, respectively; 21 °C (20 °C) for offices and commercial, educational, and public-health buildings; 20 °C for industrial buildings; 8 °C for religious and sport buildings; and 16 °C for castles[30]. A change is made between baseline and RC scenarios in the heating system capacity (from 3.23 GW, as in ref. 30 to 6.64 GW). Ref. 30 limited the capacity to a degree that the heating design temperature is only maintained for outdoor temperatures above 10 °C. This parametrises both limited heating system capacity and the fact that inhabitants limit heating during cold spells to avoid excessive energy bills. This setting is maintained for the baseline, but not for RC scenarios where this limitation is removed since it is assumed that HPs are designed with sufficient capacity and because of reduced energy consumption, inhabitants no longer try to limit energy consumption during cold spells (rebound effect; Discussion). For ACs, ref. 30 considered only very weak usage since this corresponds to their evaluation period of March 2004 to March 2005 in Toulouse when AC was nearly absent. Here, the AC design temperature for RC scenarios is modified to consider a more systematic deployment of ACs and thus enable the investigation of rebound effects in the summer. The design temperature is 26 °C (30 °C) in occupied (vacant) residential, commercial, and office buildings and 26 °C in hospitals. Buildings with other uses (agriculture, castle, industrial, religious, educational and sports) are assumed to be without ACs. Internal heat gains are due to electricity (lighting and electrical appliances) and gas (cooking and domestic hot water). A variable fraction of electric non-HVAC heat gains such as lights, electrical appliances, and domestic warm water based on the outdoor air temperature is accounted for and spans between 0.7 as reported in Table 3 of ref. 32 in winter and 1.0 in summer where gas use is assumed to be near zero. This is computed as $0.7 + 0.3 \frac{(T_t - T_{min})}{(T_{max} - T_{min})}$, with $T_t$ the outdoor dry-bulb air temperature at time $t$, and $T_{min}$ and $T_{max}$ the minimum and maximum outdoor dry-bulb air temperature over the whole period, respectively.

## Domain of investigation

Toulouse is the fourth largest city in France, with 475,438 inhabitants[54] in its main municipality. It experiences relatively mild winters with prevailing air temperatures mostly between 0 and 15 °C[55]. SURFEX-TEB simulations are conducted for a 15 × 15 km domain at 250 m horizontal resolution covering most of Toulouse (Fig. 6b) between March 2004 and February 2005 as (1) meteorological parameters for TEB from the Canopy and Aerosol Particles Interactions in TOulouse Urban Layer (CAPITOUL) campaign[56] are available for this period and (2) inventory of building energy consumption to evaluate the simulated building energy consumption is available from ref. 31. Urban morphology data such as buildings' plan area, density, and height are from ref. 57. Building construction and insulation materials are from ref. 58. Heating system types are available from the French Institute for Statistics and Economics (INSEE) at the scale of administrative areas comprising ~2000 individuals and account for the following fractions (weighted by building volume; Supplementary Fig. 2a): electric (resistive heaters) 52.3% (Supplementary Fig. 6c), natural gas 41% (Supplementary Fig. 6d), wood 4.4% (Supplementary Fig. 6e) and oil 2.3% (Supplementary Fig. 6f). It is assumed that for a given administrative area, all buildings are heated with the average share of heating system types, thus neglecting potential correlations between heating system type, and building use or size.

## Coupled online simulations

For the impact of AAHPs on meteorological conditions during a cold spell between 22 and 30 January 2005, the atmospheric model Meso-NH version 5.3—a mesoscale, anelastic, non-hydrostatic model[59,60] is used. The model is run coupled to SURFEX-TEB-MinimalDX and configured for baseline and RC3.5 scenarios. For initialisation and lateral forcing, the European Centre for Medium-Range Weather Forecasts (ECMWF) high-resolution operational analysis data are used. These data are dynamically downscaled through three intermediate nesting steps (8, 2 and 1 km) to a 250 m resolution domain covering the Toulouse metropolitan region. Models are run between 20 and 30 January 2005 but results are reported for the cold spell period between 22 and 30 January 2005 to allow for a two-day spin-up. The physical parametrisations used in the four domains are Kain and Fritsch (1990)[61] for deep convection in the outermost domain, Pergaud et al. (2009)[62] for shallow convection and dry thermals in all but the innermost domain, Bougeault and Lacarrère (1989)[63] for mixing-length calculation in all but the innermost domain where Deardorff (1980)[64] is used instead.

## Reporting summary

Further information on research design is available in the Nature Portfolio Reporting Summary linked to this article.

## Data availability

The dataset for this study is available on Zenodo at https://doi.org/10.5281/zenodo.11356004[65].

## Code availability

Software and tools are archived with a Singularity[66] image deposited on Zenodo at https://doi.org/10.5281/zenodo.11356004[65] as described in the scientific reproducibility section of ref. 22.

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

## Acknowledgements

R.S. received financial support from the French National Research Agency (Agence Nationale de la Recherche) for the project entitled "applied Modelling and urbAn Planning laws: Urban Climate and Energy (MApUCE)" with reference ANR-13-VBDU-0004. We thank Imperial College London Open Access Fund for paying this article's open access fee.

## Author contributions

Conceptualisation, data curation, investigation, methodology, software and visualisation: D.M.; Formal analysis, validation and writing—original draft preparation: D.M. and R.S.; Resources: D.M. and M.v.R.; Writing—review & editing: D.M., R.S. and M.v.R.

## Competing interests

D.M. is a strategist and consultant in the energy and HVAC sector. R.S. and M.v.R. declare no competing interests.
