## [Peer Review File · Nature Communications]

REVIEWER COMMENTS

Reviewer #1 (Remarks to the Author):

This is a well-written and very timely paper on the subject of air to air heat pumps and their effects on energy demand and the local climate.

My main comments are threefold:

1) The paper currently models the impact of different COPs ranging from 2.0-3.5 citing a now dated paper in the methodology section on real-world conditions and their impact on the COP of heat pumps. A recent meta-review of field studies (DOI: <https://doi.org/10.1016/j.joule.2022.08.015>) found that on average COPs at 0C outside air temperature were around 3.0 with some variation. It would be good for the paper to reflect this more recent analysis and data from field studies in the analysis.

2) The paper models the impact on outside air temperatures from switching from fossil fuel heating to air to air heat pumps. It is unclear to what extent the modelled changes in air temperatures result primarily from the reduction in heat generated from fossil fuels and the subsequent heat loss in buildings or the ambient heat absorbed by the heat pumps from the air. In principle, one would expect that with the heat loss being equal in both the heat pump scenario and under the status quo the primary reason for the reduced air temperature is due to the reduction in fossil fuel combustion whereas heat pumps simply transfer ambient heat to buildings which is then lost through the building fabric and leaks back into the air. It would be good for the paper to explain the driving factors here more clearly.

3) The paper cites dated EHPA analysis on the types of heat pumps sold. I encourage the author(s) to use the most recent 2022 data from EHPA. On the subject of air to air heat pumps the author(s) may also find this recent article on air to air heat pumps in the Nordic countries useful (<https://www.carbonbrief.org/guest-post-how-heat-pumps-became-a-nordic-success-story/>).

Reviewer #2 (Remarks to the Author):

The paper explores the potential effects of transitioning entirely to air-to-air heat pumps and their performance across several building and weather conditions in Toulouse, France. In addition to the main results, it provides an extensive discussion on the complexities of electrification's environmental and economic trade-offs, underscoring the importance of regional analyses and broader policy considerations.

- In the reviewer's opinion, the paper is difficult to understand in some parts AND the following changes WOULD ease ITS understanding:

- Specifying the outline of the paper;
- Moving the modelling approach section to the methodology;
- Naming the introduction and results sections;
- adding subsections to the methodology;
- Furthermore, further review of previous studies should be added in the introduction highlighting the novelty of the present paper.
- The results are complete but they must be improved, adding further explanation of the figures and tables (such as the meaning of the intervals of the energy use plots, the surface air temperature section, etc.).
- The discussion is correct but the conclusions of the present paper should be better explained in order to highlight its contribution, which is not entirely clear for the reviewer.
- Finally, the methodology should be improved in order to understand how the results of the paper were obtained.
- There are some typos that should be corrected, labels in figure 2b, 3b, and 4b are missing and figure 1 could be deleted since it is out of the scope of the paper.

Reviewers' comments

Reviewer #1 (Remarks to the Author):

This is a well-written and very timely paper on the subject of air to air heat pumps and their effects on energy demand and the local climate.

Thank you for taking the time to provide such a careful revision of our submission. We addressed all your comments. Our detailed answers are given below.

My main comments are threefold:

1) The paper currently models the impact of different COPs ranging from 2.0-3.5 citing a now dated paper in the methodology section on real-world conditions and their impact on the COP of heat pumps. A recent meta-review of field studies (DOI: <https://doi.org/10.1016/j.joule.2022.08.015>) found that on average COPs at 0C outside air temperature were around 3.0 with some variation. It would be good for the paper to reflect this more recent analysis and data from field studies in the analysis.

Thanks, we have altered the number of rated COP (RC) scenarios to 5 and changed the range. Specifically, we have removed the RC@2.0 scenario as too low and added RC@4.0 and RC@4.5 scenarios. The average scenario is taken as RC@3.5 (Figure RR1) which is comparable to the trendline shown in Gibb et al., 2023's Fig. 2.

Figure RR1. Results from multiple MinimalDX (Meyer & Raustad, 2020) simulations using the following input parameters: Outdoor dry-bulb air temperature between -5 °C and 15 °C, outdoor relative humidity 50%, outdoor air pressure 101325 Pa, indoor air temperature: 18 °C, indoor relative humidity 50%, rated COP 3.5, rated total capacity 3 kW, sensible heating load 3 kW, defrost power 600 W. Blue scatters show each configuration. The trendline is shown with a dashed red line.

2) The paper models the impact on outside air temperatures from switching from fossil fuel heating to air to air heat pumps. It is unclear to what extent the modelled changes in air temperatures result primarily from the reduction in heat generated from fossil fuels and the subsequent heat loss in buildings or the ambient heat absorbed by the heat pumps from the air. In principle, one would expect that with the heat loss being equal in both the heat pump scenario and under the status quo the primary reason for the reduced air temperature is due to the reduction in fossil fuel combustion whereas heat pumps simply transfer ambient heat to buildings which is then lost through the building fabric and leaks back into the air. It would be good for the paper to explain the driving factors here more clearly.

We agree with the reviewer. In the present study, we assume that the building envelope remains the same for the baseline and heat pump scenarios. The heating design temperature also remains the same, the only exception is when we consider rebound effects. Therefore, the leakage of heat through the building envelope is very similar for both scenarios (only slightly differing due to feedback like the modification of outdoor temperature due to the heat pumps). The outdoor air temperature reduction is therefore due to the replacement of fossil-fuel combustion and electric resistive heating, which is an external heat source, by the local heat source (outdoor air). We now explain this better in the introduction.

3) The paper cites dated EHPA analysis on the types of heat pumps sold. I encourage the author(s) to use the most recent 2022 data from EHPA. On the subject of air to air heat pumps the author(s) may also find this recent article on air to air heat pumps in the Nordic countries useful (<https://www.carbonbrief.org/guest-post-how-heat-pumps-became-a-nordic-success-story/>).

Many thanks. We have added more recent references as suggested. We have further updated Eurostat and IEA references in the introduction with the most up-to-date statistics.

Reviewer #2 (Remarks to the Author):

The paper explores the potential effects of transitioning entirely to air-to-air heat pumps and their performance across several building and weather conditions in Toulouse, France. In addition to the main results, it provides an extensive discussion on the complexities of electrification's environmental and economic trade-offs, underscoring the importance of regional analyses and broader policy considerations.

Thank you for taking the time to provide such a careful revision of our submission. We addressed all your comments—detailed answers are given below.

- In the reviewers opinion, the paper is difficult to understand in some parts AND the following changes WOULD ease ITS understanding:
- Specifying the outline of the paper;

We understand the importance of guiding the reader through the narrative and structure of our work. We have substantially reworded the introduction and other sections throughout to improve clarity. We have included a paragraph under “Results” -> “Energy consumption”, which should improve clarity on the section’s content. We have decided to adhere to the stylistic conventions of Nature Communications, which, as we understand it, does not typically include detailed outlines of the paper’s structure within the introduction section. We believe that the current structure of our introduction, coupled with clearly defined section headings throughout the manuscript, effectively guides the reader through our research findings and their implications.

- Moving the modelling approach section to the methodology

The “modelling approach” section has been moved to the methodology section and been renamed to “Simulated AAHP scenarios”, since this represents better its content.

- Naming the introduction and results sections
Done.

- adding subsections to the methodology;

More subsections have been added to the methodology to make it easier to read. Additionally, subsections have been added to the discussion and conclusion.

- Furthermore, further review of previous studies should be added in the introduction highlighting the novelty of the present paper.

We have rephrased the introduction and added review of more studies, accordingly, highlighting the novelty of our study. We have also added more references (Reviewer 1’s Comment 3).

- The results are complete but they must be improved, adding further explanation of the figures and tables (such as the meaning of the intervals of the energy use plots, the surface air temperature section, etc.).

We agree that some explanations were short, therefore we have added more.

- The discussion is correct, but the conclusions of the present paper should be better explained in order to highlight its contribution, which is not entirely clear for the reviewer.

The discussion and conclusion section has been revised and the contribution of this paper is now more clearly stated in a dedicated subsection.

- Finally, the methodology should be improved in order to understand how the results of the paper were obtained.

More subsection headings were added to the methodology section and the order of the subsections has been changed such as to make it easier to follow.

- There are some typos that should be corrected, labels in figure 2b, 3b, and 4b are missing and figure 1 could be deleted since it is out of the scope of the paper.

Thank you. We have fixed all the typos and substantially reworked the text as suggested in your earlier points. We hope that figure 1 is better placed.

References

- Gibb, D., Rosenow, J., Lowes, R., & Hewitt, N. J. (2023). Coming in from the cold: Heat pump efficiency at low temperatures. *Joule*, 7(9), 1939–1942. <https://doi.org/10.1016/j.joule.2023.08.005>
- Meyer, D., & Raustad, R. (2020). MinimalDX. *Zenodo*. <https://doi.org/10.5281/zenodo.3892452>

REVIEWER COMMENTS

Reviewer #1 (Remarks to the Author):

The authors have addressed all of my comments. My only additional comment is that the paper incorrectly states that air to air are dominant in Europe. This is not true for the whole of the EU as the 2022 data shows (<https://heatpumpingtechnologies.org/growth-record-for-european-heat-pump-market-2022-best-year-ever/>). It is true however in the Nordic countries as established by reference [13].

If that change is made in the text I would be happy to accept publication now.

Reviewer #2 (Remarks to the Author):

The comments of both reviewers have been correctly added, no further changes are needed.

Reviewers' comments

Reviewer #1 (Remarks to the Author):

The authors have addressed all of my comments. My only additional comment is that the paper incorrectly states that air to air are dominant in Europe. This is not true for the whole of the EU as the 2022 data shows (<https://heatpumpingtechnologies.org/growth-record-for-european-heat-pump-market-2022-best-year-ever/>). It is true however in the Nordic countries as established by reference [13].

If that change is made in the text I would be happy to accept publication now.

Thank you. We have simplified the sentence and addressed the issue. In addition, we have made the following minor changes:

- Estimated CO₂ emissions values are now taken directly from our baseline simulation—please see new Supplementary Fig. 8 and “Impact on CO₂ emissions and urban climate” and “Rebound effect and radiative forcing” sections.
- Estimated CO₂ equivalent emissions values are reported for R32 only as this supersedes R410-A. Please see “Rebound effect and radiative forcing” section.
- Fixed typos in Table 1 Heating and Cooling EEC_{centre} for RC@4.0 and RC@4.5.
- Other minor copyediting changes such as revised title and updated ordering and labelling of supplementary figures.

Reviewer #2 (Remarks to the Author):

The comments of both reviewers have been correctly added, no further changes are needed.

Thank you.